The structure of people’s hair

Yang Fei-Chi
Zhang Yuchen
Rheinstädter Maikel C. rheinstadter@mcmaster.ca
Department of Physics and Astronomy, McMaster University , Hamilton, Ontario , Canada
Karttunen Mikko
Electronic publication date: 2014 Oct 14
Publication date: 2014
Volume: 2
Electronic Location ID: e619
Received 2014 Aug 7; Accepted 2014 Sep 22
Copyright: © 2014 Yang et al.
Copyright year: 2014
Copyright holder: Yang et al.
License: This is an open access article distributed under the terms of the Creative Commons Attribution License, which permits unrestricted use, distribution, reproduction and adaptation in any medium and for any purpose provided that it is properly attributed. For attribution, the original author(s), title, publication source (PeerJ) and either DOI or URL of the article must be cited.
License URL: https://creativecommons.org/licenses/by/4.0/

Keywords: Human hair, Molecular structure, X-ray diffraction, Keratin, Intermediate filament, Coiled-coil proteins, Alpha helix, Cell membrane complex

Funding: Natural Sciences and Engineering Research Council of Canada (NSERC) National Research Council Canada (NRC) Canada Foundation for Innovation (CFI) Ontario Ministry of Economic Development and Innovation This research was funded by the Natural Sciences and Engineering Research Council of Canada (NSERC), the National Research Council Canada (NRC), the Canada Foundation for Innovation (CFI) and the Ontario Ministry of Economic Development and Innovation. MCR is the recipient of an Early Researcher Award of the Province of Ontario. The funders had no role in study design, data collection and analysis, decision to publish, or preparation of the manuscript.

==============================
Hair is a filamentous biomaterial consisting mainly of proteins in particular keratin. The structure of human hair is well known: the medulla is a loosely packed, disordered region near the centre of the hair surrounded by the cortex, which contains the major part of the fibre mass, mainly consisting of keratin proteins and structural lipids. The cortex is surrounded by the cuticle, a layer of dead, overlapping cells forming a protective layer around the hair. The corresponding structures have been studied extensively using a variety of different techniques, such as light, electron and atomic force microscopes, and also X-ray diffraction. We were interested in the question how much the molecular hair structure differs from person to person, between male and female hair, hair of different appearances such as colour and waviness. We included hair from parent and child, identical and fraternal twins in the study to see if genetically similar hair would show similar structural features.

The molecular structure of the hair samples was studied using high-resolution X-ray diffraction, which covers length scales from molecules up to the organization of secondary structures. Signals due to the coiled-coil phase of α-helical keratin proteins, intermediate keratin filaments in the cortex and from the lipid layers in the cell membrane complex were observed in the specimen of all individuals, with very small deviations. Despite the relatively small number of individuals (12) included in this study, some conclusions can be drawn. While the general features were observed in all individuals and the corresponding molecular structures were almost identical, additional signals were observed in some specimen and assigned to different types of lipids in the cell membrane complex. Genetics seem to play a role in this composition as identical patterns were observed in hair from father and daughter and identical twins, however, not for fraternal twins. Identification and characterization of these features is an important step towards the detection of abnormalities in the molecular structure of hair as a potential diagnostic tool for certain diseases.

Introduction

Human scalp hair is a bio-synthesized material that has a complex internal structure. The adult human hair is around 20–180 µm in width, and generally grows to a length of approximately 90 cm. It consists of many layers including the cuticle, the cortex and the medulla. These layers are bound together by the cell membrane complex (Robbins, 2012).

The structure of human hair is well known and in particular X-ray diffraction revealed details of molecular structure and organization within hair (Fraser et al., 1986; Briki et al., 2000; Busson, Engstrom & Doucet, 1999; Randebrook, 1964; Fraser, MacRae & Rogers, 1962; Kreplak et al., 2001b; Wilk, James & Amemiya, 1995; Pauling & Corey, 1951; Ohta et al., 2005; Astbury & Street, 1932; Astbury & Woods, 1934; Astbury & Sisson, 1935; Franbourg et al., 2003; Rafik, Doucet & Briki, 2004; James et al., 1999; Veronica & Amemiya, 1998; Briki et al., 1999; James, 2001). In particular microbeam small angle X-ray scattering techniques enables the determination of hair structure with a high spatial resolution (Iida & Noma, 1993; Busson, Engstrom & Doucet, 1999; Kreplak et al., 2001b; Ohta et al., 2005; Kajiura et al., 2006). It is a long-standing question whether changes in the molecular structure of nail or hair can be related to certain diseases and potentially be used as a diagnostic tool. Such a technique would in particular be interesting and relevant as simple, non-invasive screening method for cancer (James et al., 1999; Briki et al., 1999; James, 2001). Abnormal kinky hair is, for instance, characteristic of giant axonal neuropathy (Berg, Rosenberg & Asbury, 1972).

The purpose of this study is to use X-ray diffraction to analyze the structure of human scalp hair for individuals with differing characteristics. The 12 individuals in this study include hair from men and women and hair of different colour and appearance, such as straight, wavy and curly. In addition to appearance, the study also includes hair from a father and daughter, a pair of identical and a pair of fraternal twins to include genetic similarities. All hair was collected from healthy individuals and care was taken that the hair was not permed or dyed before the experiments.

Signals due to the coiled-coil organization of α-helical keratin proteins and intermediate filaments in the cortex, and lipids in the cell membrane complex were observed in the hair of all individuals. While these general features occur independent of gender or appearance of the hair with a very small standard deviation in the underlying molecular dimensions, we find significant differences between individuals in the composition of the plasma membrane in the cell membrane complex. Genetics appear to be the most important factor that determines membrane composition, as no or little differences were observed in genetically related hair samples, rather than external factors such as nutrition or hair care products.

Properties of human hair

The cuticle is the outermost layer formed by flat overlapping cells in a scale-like formation (Robbins, 2012). These cells are approximately 0.5 µm thick, 45–60 µm long and found at 6–7 µm intervals (Robbins, 2012). The outermost layer of the cuticle, the epicuticle, is a lipo-protein membrane that is estimated to be 10–14 nm thick (Swift & Smith, 2001). Beneath that is the A layer with a high cysteine content and a thickness of 50–100 nm, the exocuticle with again a high cysteine content and a highly variable thickness ranging from 50 to 300 nm, and the endocuticle with a low cysteine content and a thickness also ranging from 50 to 300 nm.

The majority of hair fibre is the cortex which contains spindle shaped cells that lie parallel along the fibre axis. These cortical cells were found to be approximately 1–6 µm in diameter and 50–100 µm in length (Randebrook, 1964). In wool fibres as well as human hair, the cortical cells were observed to be divided into different regions termed orthocortex, paracortex and mesocortex (Mercer, 1953). The difference in distribution of these cell types is an important factor for determining the curvature of the hair fibre (Kajiura et al., 2006). In particular, straight hair tends to have symmetrical distribution of the ortho- and paracortices whereas curly hair tends to have a non-symmetrical distribution of these cortical cells (Kajiura et al., 2006). Most of the cortical cells are composed of a protein known as keratin (Robbins, 2012).

At the molecular level, keratin is a helical protein (Pauling & Corey, 1950). There are two types of keratin fibres that exist in hair: type I with acidic amino acid residues and type II with basic amino residues. One strand of type I fibre and one strand of type II fibre spiral together to form coiled-coil dimers. In turn, these dimers coil together in an antiparallel manner to form tetramers (Crewther et al., 1983; Fraser et al., 1988).

When tetramers are connected from head to tail, they are known as protofilaments (Robbins, 2012). These tetramers or protofilaments are believed to interact together to form a single intermediate filament which is approximately 75–90 Å in diameter. The current model of an intermediate filament was proposed in the 1980’s and it involves 7 protofilaments surrounding a single core protofilament (Robbins, 2012; Fraser et al., 1988). The intermediate filaments then aggregate together to form macro-filaments with a diameter of 1000 to 4000 Å (Robbins, 2012; Randebrook, 1964). Between the intermediate filaments is a matrix consisting of keratin associated proteins, which are irregular in structure. The macro-fibrils consisting of intermediate filaments and the surrounding matrix are the basic units of the cortical cell.

The cell membrane complex is the material that glues hair cells together. There exist various types of cell membrane complexes: cuticle–cuticle, cuticle–cortex and cortex–cotex depending on the location (Robbins, 2012). The general membrane structure is one 15 nm proteinous delta layer sandwiched by two 5 nm lipid beta layers (Rogers, 1959). Much speculation still exist regarding the precise structure of the beta and delta layers. However, it has been determined that 18-methyl eicosanoic acid, a covalently bound fatty acid, exists in the upper beta layer in the cuticle–cuticle but not in cortex–cortex membranes (Ward & Lundgren, 1954). In fact, most of the fatty acids in beta layers of membranes in the cuticle–cuticle are covalently bound and most of the fatty acids in the beta layers of cortex–cortex are non-covalently bound (Robbins, 2012). Further evidence suggests that the fatty acids in cuticle–cuticle membranes are organized in a monolayer whereas the fatty acids in cortex–cortex cell membranes are bilayers (Robbins, 2012). The cuticle–cortex cell membrane complex is then a mixture of the two, with the side facing the cuticle similar to cuticle–cuticle membranes and the side facing the cortex similar to cortex–cortex membranes (Robbins, 2012).

Materials and Methods

Preparation of hair samples

This research was approved by the Hamilton Integrated Research Ethics Board (HIREB) under approval number 14-474-T. Written consent was obtained from all participating individuals. Scalp hair samples were gathered from 12 adults of various age, gender, ethnicities, hair colour and hair curvature. It is of interest to note that there are 3 pairs of study participants with genetic relations including a father and daughter, fraternal twins and identical twins. Characteristics of the samples are listed in Table 1.

Table 1 List of all hair samples in this study.

The individuals include men and women and hair of different appearance, such as thickness, colour and waviness, and also genetically related hair samples from a father and daughter, a pair of identical and a pair of fraternal twins. Labeling agrees with the data shown in Fig. 1.

Subject	Gender	Diameter(µm) ± SD	Colour	Appearance	Special comment	
1	F	30 ± 3	light blonde	straight	daughter	
2	M	49 ± 5	brown/grey	curly	father	
3	F	74 ± 7	black	wavy	–	
4	M	50 ± 5	light brown	curly	–	
5	F	49 ± 5	blonde	curly	–	
6	F	43 ± 4	light brown	straight	–	
7	F	61 ± 6	light brown	wavy	–	
8	F	49 ± 5	black	wavy	–	
9	F	31 ± 3	blonde	wavy	identical twin	
10	F	66 ± 7	black	straight	fraternal twin	
11	F	69 ± 7	black	straight	fraternal twin	
12	F	48 ± 5	blonde	curled	identical twin	

The hair samples gathered were cut into strands around 3 cm long. Care was taken to not stretch or deform the hair strands during this process. For each subject, around 10 strands were taped onto a flexible cardboard apparatus as shown in Fig. 2. The cut-out at the middle of the apparatus is where scattering occurs on the hair sample. The cardboard apparatus is then mounted vertically onto the loading plate of the Biological Large Angle Diffraction Experiment (BLADE) using sticky putty as shown in Fig. 2. All hair samples were measured at room temperature and humidity of 22 °C and 50% RH.

X-ray diffraction experiment

X-ray diffraction data was obtained using the Biological Large Angle Diffraction Experiment (BLADE) in the Laboratory for Membrane and Protein Dynamics at McMaster University. BLADE uses a 9 kW (45 kV, 200 mA) CuKα Rigaku Smartlab rotating anode at a wavelength of 1.5418 Å. Focusing multi-layer optics provided a high intensity parallel beam with monochromatic X-ray intensities up to 1010 counts/(s × mm2) at the sample position. In order to maximize the scattered intensity, the hair strands were aligned parallel to the parallel beam for maximum illumination. The slits were set such that about 15 mm of the hair strands were illuminated with a width of about 100 µm. The effect of this particular beam geometry is seen in the 2-dimensional data in Fig. 1: while it produces a high resolution along the equator, the main beam is significantly smeared out in the qz-direction up to qz-values of about 0.5 Å-1, limiting the maximum observable length scale to about 13 Å.

Figure 1 Two-dimensional X-ray data of all 12 subjects.

The hair strands were oriented with the long axis of the hair parallel to the vertical z-axis. The (q∥, qz)-range shown was determined in preliminary experiments to cover the features observable by X-ray diffraction. The measurements cover length scales from about 3–90 Å to study features from the coiled-coil α-keratin phase, keratin intermediate filaments in the cortex, and the membrane layer in the membrane complex. While common features can easily be identified in the 2D plots, subtle differences are visible, which are discussed in detail in the text.

Figure 2 The apparatus used to mount the hair strands in the experiment.

The cardboard apparatus is mounted vertically onto the loading plate of the Biological Large Angle Diffraction Experiment (BLADE) using sticky putty.

Figure 3 Schematics of the X-ray setup and example X-ray data.

The hair strands were oriented in the X-ray diffractometer with their long axis along qz. Two-dimensional X-ray data were measured for each specimen covering distances from about 3–90 Å including signals from the coiled-coil α-keratin phase, the intermediate fibrils in the cortex and from the cell membrane complex. The 2-dimensional data were integrated and converted into line scans and fit for a quantitative analysis.

The diffracted intensity was collected using a point detector. Slits and collimators were installed between X-ray optics and sample, and between sample and detector, respectively. By aligning the hair strands in the X-ray diffractometer, the molecular structure along the fibre direction and perpendicular to the fibres could be determined. We refer to these components of the total scattering vector, Q→, as qz and q‖, respectively, in the following. An illustration of qz and q‖ orientations is shown in Fig. 3. The result of an X-ray experiment is a 2-dimensional intensity map of a large area of the reciprocal space of −2.5 Å−1 < qz < 2.5 Å−1 and −2.5 Å−1 < q‖ < 2.5 Å−1. The corresponding real-space length scales are determined by d = 2π/|Q| and cover length scales from about 3 to 90 Å, incorporating typical molecular dimensions and distances for secondary protein and lipid structures.

Integration of the 2-dimensional data was performed using Matlab, MathWorks. By adding up the peak intensities along the qz and the q‖ directions, 1-dimensional data along each of the two directions were produced. The qz intensity was integrated azimuthally for an angle of 25 degrees over the meridian. The q‖ intensity was integrated azimuthally for an angle of 25 degrees over the equator, as depicted in Fig. 3.

The fitting process is performed on both the 1-dimensional qz and the q‖ data produced from integration. Distinguishable peaks were observed and fitted with the least numbers of Lorentzian peak functions with an exponential decay background of the form (a⋅qb + c) in the first run. Initial Parameters were chosen based on the observed positions, widths and heights of the peaks and free to move through the entire q-range. The criterion for the final parameters was to minimize the mean square of the difference between data intensity and the fitted intensity. If the fitted intensity cannot conform to the shape of the data intensity, more peaks will be added in the following runs until a good fit is acquired. This process was repeated for all 12 subjects and performed with little or no consultation of previous fittings to minimize bias.

As for the SAXS data, Gaussian functions are used instead. We note that the use of optical components in the beam path has an impact on the shape of the observed Bragg peaks: instead of Lorentzian or Bessel peak functions, Gaussian peak profiles were found to best describe the SAXS peaks. The fitting process was the same as mentioned before: three Gaussians were fitted to the SAXS data using free-to-move parameters and an exponential decay background. However, for some subjects, the third peak was noisy and the least mean square logarithm could not reach a good fit and hence the data was fitted with two Gaussians, only.

Results

A total of 12 adult subjects participated in this study. Details of gender and appearance of the hair strands are listed in Table 1. About 10 strands were cut from the scalp, glued onto a sample holder and aligned in the X-ray diffractometer. The resulting 2-dimensional X-ray intensity maps of the reciprocal space reveal exquisite details of the molecular structure of human scalp hair, as presented in Fig. 1. The hair strands were oriented with the long axis of the hair parallel to the vertical z-axis. The displayed (qz, q‖)-range was determined to cover the length scales of the features of interest in preliminary experiments.

Figure 4 The hierarchical structure of hair in the cortex and cuticle.

The main component of the cortex is a keratin coiled-coil protein phase. The proteins form intermediate filaments, which then organize into larger and larger fibres. The hair is surrounded by the cuticle, a dead cell layer. The common features observed in the X-ray data of all specimens are signals related to the coiled-coil keratin phase and the formation of intermediate filaments in the cortex, and the cell membrane complex. Signal assignment and corresponding length scales are shown in the figure.

The data in Fig. 1 show a distinct non-isotropic distribution of the diffracted intensity with pronounced and well defined intensities along the long axis of the hair and in the equatorial plane (the qz and q‖-axes, respectively), indicative of a high degree of molecular order in the hair strands. Some features were common in all specimens and assigned to certain molecular components, as explained in the next section.

Assignment of common scattering signals

Coiled-coil protein phase in the cortex

The keratin proteins in the cortex are known to organize in bundles whose structures are dominated by α-helical coiled-coils (Pauling & Corey, 1950; Pinto et al., 2014; Yang et al., 2014). The main features of this pattern are a ∼9.5 Å (corresponding to q‖ ∼ 0.6 Å−1) equatorial reflection corresponding to the spacing between adjacent coiled-coils and a ∼5.0 Å meridional reflection (corresponding to qz ∼ 1.25 Å−1) corresponding to the superhelical structure of α-helices twisting around each other within coiled-coils (Crick, 1952; Cohen & Parry, 1994; Lupas & Gruber, 2005). As displayed in Fig. 4, these signals were observed in the X-ray data in all specimen and assigned to the coiled-coil protein phase. We note that these peaks are related to generic α-helical coil structures of monomeric proteins, and not specific to a certain type of protein.

Lipids in the cell membrane complex

The cell membrane complex mainly consists of lipid mono- and bilayers. The corresponding scattering features correspond to a lamellar periodicity of about 45 Å, and rings at spacings of about 4.3 Å, characteristic of the order within the layers (Busson, Engstrom & Doucet, 1999). Both these features are observed in the 2-dimensional X-ray data of all individuals in Fig. 1, as a ring-like scattering intensity at q-values of ∼0.1 Å-1 and a broad, ring-like scattering at ∼1.5 Å-1 as a result of the lipid order within the membrane layers. The corresponding diffraction signal has a maximum on the qz-axis, indicating a preferential orientation of the membrane plane parallel to the surface of the hair.

Intermediate filaments in the cortex

The keratin coils organize into intermediate filaments whose structure and packing in the plane of the hair result in additional scattering signals. The packing of these fibrils by bundling into macro-fibrils is characterized by X-ray diffraction pattern by three equatorial spots located at about 90, 45 and 27 Å (Busson, Engstrom & Doucet, 1999). The corresponding signals are observed in the 2-dimensional data in Fig. 1. The exact position of the features is, however, best determined in small angle diffraction experiments (SAXS), which offer a drastically improved resolution, and will be shown below. We note that the axial packing of coiled-coils within keratin filaments in hair gives rise to a number of fine arcs along the meridian (z). The typically observed signal on the meridian at 67 Å, which arises from the axial stagger between molecules along the microfibril (Briki et al., 2000; Rafik, Doucet & Briki, 2004), could not be observed in our experiments due to the relaxed resolution of the parallel beam in this direction. While the features observed in scattering experiments are well known, the molecular architecture of the intermediate filaments is still under discussion (Rafik, Doucet & Briki, 2004). Supercoiled coiled-coils or models that involve straight dimers with different numbers of coils are being discussed.

The three features above were observed in all individuals in Fig. 1. The underlying molecular structures will be quantitatively analyzed in the next section (Quantitative analysis of scattering results). We note that additional features are seen in some of the measurements in Fig. 1, mainly in the broad membrane ring at around 1.5 Å-1 which indicates a difference in molecular composition of the cell membrane complex between individuals. We will come back to these differences in the Discussion.

Quantitative analysis of scattering results

In order to quantitatively determine the position of the corresponding scattering features, the 2-dimensional data for all 12 individuals were integrated in the equatorial plane (q‖-axis) of the hair fibres, and along the hair fibres (qz-axis). The resulting plots are shown in Fig. 5. In the direction along the hair fibre axis (qz), there are two major peaks that were consistent among all subjects, one narrow peak around 5.0 Å and one broader peak around 4.3 Å.

Figure 5 Integration of the 2-dimensional scattering data in Fig. 1 in the equatorial plane (q‖) (A), and along the axis of the hairs (qz) (C), respectively, for all subjects. The two signals present in all individuals in the equatorial plane (q‖) correspond to the distance between two coiled coils of 9.5 Å and between two lipid tails in the cell membrane cortex of 4.3 Å. The common meridional signal along the long axis of the hair (qz) at 5 Å corresponds corresponds to the α-helices twisting around each other within coiled-coils. Average values and standard deviations are in (B).

In the direction perpendicular to the hair fibre axis (q‖), there are also two major peaks consistent among all subjects, one narrow peak around 9.5 Å and one broad peak around 4.3 Å. The total scattering profile was well fit by two Lorentzian peak profiles (and a background), whose positions is plotted in Fig. 5. The signals at 5.0 Å and 9.5 Å are in excellent agreement with signals reported from coiled-coil keratin proteins (Pauling & Corey, 1950), as depicted in the Figure. The broad signal at about 4.3 Å present in both directions is due to the ring-like scattering from the lipids in the membrane component. As plotted in Fig. 5, there is a narrow distribution of the corresponding length scales with standard deviations of 9.51 ± 0.07 Å and 5.00 ± 0.02 Å for the keratin coiled-coils and 4.28 ± 0.08 Å for the membrane signal, indicating that the common features observed in all individuals are well defined with little spread in the corresponding molecular dimensions.

Figure 6 Diffraction features at small scattering angles.

The small q‖-range is shown in magnification in (A). The specimen of most individuals showed 3 distinct reflections at ∼90 Å, 46.5 Å and 27 Å, related to the properties of intermediate keratin filaments (B).

Due to the large length scales involved, the signals from intermediate filaments occur at small scattering vectors, shown in Fig. 6. The Small Angle X-ray Scattering (SAXS) profile was well fit with three Gaussian peaks at 90 Å, 45 Å, and 27 Å. We note that the third peak was not observed in all hair samples. The corresponding peak positions and distributions are shown in the figure. The 90 Å peak has been reported early in the literature as the distance between intermediate filaments in human hair. As further elaborated by Rafik, Doucet & Briki (2004), these peaks correspond to the radial structures of the intermediate filaments and can be well-simulated by assuming parallel tetramers formed by 2 coiled-coils with a slight disorder in positions and orientations, as depicted in the figure. Also here, the standard deviations of 90 ± 2 Å, 47 ± 2 Å, 27 ± 1 Å, as shown in the figure, are small, indicating that the organization of the intermediate filaments on the nanoscale varies very little between different individuals.

Discussion

All hair used in this study was in its native state, collected from healthy individuals and not chemically treated prior to the experiments. However, all individuals regularly used shampoos for cleaning and additional products such as conditioners, wax and gel. These products function primarily at or near the fiber surface to remove dirt from the hair surface, for instance, and do not seem to have an impact on the internal keratin structure, as will be discussed below.

An abnormal signal was previously reported by James et al. (1999) in hair samples of patients with breast cancer. Such an approach is quite intriguing, as scanning of hair samples could be used as easy, inexpensive and non-invasive screening techniques in the diagnosis of cancer. James et al. (1999) observed a ring-like signal at 44.4 Å, at the position of the lamellar plasma membrane signal, and assigned this signal to the presence of breast cancer. The analysis and assignment was questioned later on by Briki et al. (1999) and Howell et al. (2000), who observed this feature in healthy and cancer patients in equal measure. The ring-like 45 Å signal is also present in the data for all individuals included in our study, such that a relation to breast cancer can most likely be excluded.

General structural features from the X-ray experiments

From the 2-dimensional X-ray data in Figs. 1 and 4, and the analysis in Figs. 5 and 6, we identify three features present in all individuals. These signals are related to the coiled-coils arrangement of the keratin proteins in the cortex, the formation of intermediate filaments in the cortex, and lipids in the cell membrane complex of the hair. Statistical analysis of the corresponding molecular dimensions revealed a rather small distribution between different individuals. These general properties of human hair are observed in all hair independent of gender, colour or optical appearance of the hair (as listed in Table 1) within the number of individuals included in this study.

Figure 7 Comparison between hair samples.

(A) shows a comparison between individuals 3 and 4. While the two specimens both show the general features, differences are observed in the region of signal from the cell membrane complex. (B) Comparison between individuals 1 and 2, father and daughter. The data in (C) (individuals 9 and 12) are from identical twins. Data in (D) was taken from fraternal twins (individuals 10 and 11). While different individuals in general show different membrane patterns (A), features in (B) and (C) perfectly agree. Fraternal twins show slight differences in their pattern in (D).

Differences in the X-ray data between individuals were observed in the wide angle region (WAXS) of the 2-dimensional data in Fig. 1, related to properties of the membrane component. Figure 7A shows a comparison between individual 3 and 4 to illustrate the effect. For an easy comparison, the original data were cut in half and recombined, such that the left half depicts individual 3, and the right half individual 4. While signals from the coiled-coil protein phase, the diffuse, ring-like intensity from lipids in the cell membrane complex and the small angle signals due to the formation of intermediate filaments are observed in both individuals, additional signals occur in Subject 3 around the position of the membrane-ring. Almost identical patterns are observed in Figs. 7B and 7C, while differences are seen in Fig. 7D; this will be discussed in detail below.

The additional signals observed between about 1.34 Å-1 and 1.63 Å-1 can be assigned to fatty acids located within the plasma membrane of the cell membrane complex. The position of these lipids inside the hair was determined by synchrotron infrared microspectroscopy (Kreplak et al., 2001a) detecting the corresponding CH2 and CH3 bands. The lipid component of the cell membrane complex consists of three major classes of lipids: glycerolipids (mainly phospholipids), sterols and sphingolipids (Furt, Simon-Plas & Mongrand, 2011). The most abundant lipid species are referred to as structural lipids up to 80% of which are phosphocholine (PC) and phosphoethanolamine (PE) phospholipids.

The position and width of the broad, ring-like intensity observed in all specimens in Fig. 1 agree well with lipid correlation peaks reported from single and multi-component phospholipid fluid lipid membranes (Kučerka et al., 2005; Petrache et al., 1998; Kuč, Tristram-Nagle & Nagle, 2006; Rheinstädter et al., 2004; Rheinstädter, Seydel & Salditt, 2007; Rheinstädter et al., 2008; Pan et al., 2008; Schneggenburger et al., 2011; Harroun et al., 1999) and diffraction observed in plasma membranes (Welti et al., 1981; Poinapen et al., 2013). The broad correlation peak is the tell-tale sign of a fluid-like, disordered membrane structure. It is related to the packing of the lipid tails in the hydrophobic membrane core, where the lipid acyl chains form a densely packed structure with hexagonal symmetry (planar group p6) (Armstrong et al., 2013). The distance between two acyl tails is determined to be aT=4π/3qT (Mills et al., 2008; Barrett et al., 2012; Barrett et al., 2013), where qT is the position of the membrane correlation peak. The average nearest-neighbour distance between two lipid tails is calculated from the peak position to 4.97 Å. We note that the intensity of the disordered membrane component is not distributed isotropically on a circle, which would be indicative of a non-oriented, isotropic membrane phase. The corresponding scattering signal has a maximum along the qz-axis, indicative that most of the membranes are aligned parallel to the hair surface.

The additional narrow components in Fig. 1 between about 1.34 Å-1 and 1.63 Å-1, which are observed in some hair samples, agree with structural features reported in lipid membranes of different composition. A correlation peak at ∼1.5 Å-1 was found in the gel phase of saturated phospholipid membranes, such as DMPC (Dimyristoyl-sn-glycero-3-phosphocholine) and DPPC (Dipalmitoyl-sn-glycero-3-phosphocholine) (Tristram-Nagle et al., 2002; Katsaras et al., 1995; Rheinstädter et al., 2004). Unsaturated lipids were reported to order in a structure with slightly larger nearest neighbour tail distances, leading to an acyl-chain correlation peak at ∼1.3 Å-1, as reported for DOPC and POPC (Mills et al., 2009), for instance. Lipids, such as Dimyristoylphosphatidylethanolamine (DMPE) and the charged DMPS (Dimyristoyl-sn-glycero-3-phosphoserine) with smaller head groups were reported to order in more densely packed structures (Rappolt & Rapp, 1996). The corresponding acyl chain correlation peaks were observed at Q values of ∼ 1.65 Å−1. The observed differences in the X-ray diffraction patterns between different individuals can, therefore, most likely be assigned to differences in the molecular composition of the plasma membrane in the cell membrane complex. Genetics plays an important role in this composition.

Genetic similarity

Some subjects have genetic relations within the subject pool. In particular, Subject 1 and 2 are daughter and father, Subjects 10 and 11 are fraternal twins, and Subjects 9 and 12 are identical twins. The corresponding diffraction data are shown in Figs. 7B, 7C and 7D. While in general, the diffraction patterns in the membrane region were found to be different (as demonstrated in Fig. 7A), the genetically similar hair of father and daughter and identical twins show identical patterns within the resolution of our experiment.

It is interesting to note that differences are observed for the fraternal twins in Fig. 7D. This finding is in agreement with the expectation that individuals with similar genetics would share similar physical traits such as hair structure. Identical or monozygotic twins originate from one zygote during embryonic development, and they share 100% of their genetic material. Fraternal or dizygotic twins develop from the fertilization of two different eggs and they only share 50% of their DNA on average (Nussbaum et al., 2007).

As expected, the identical twin pair shows almost identical hair structures whereas the fraternal pair exhibits distinct differences. Offspring receive half of their chromosomes from each parent, thus the genetic similarity between the parent and child pair is roughly the same as fraternal twins (Creasy et al., 2013). It is, therefore, surprising that the father and daughter pair share significantly more similarities than the pair of fraternal twins. This can be attributed to the fact that the expression of a complex trait such as hair structure would depend on the inheritance pattern of many phenotype-determining genes, such as whether they are dominant or recessive traits. Genetic similarity does not guarantee identical hair structure and similarly, genetic variability does not guarantee differences. While we can report this finding, the small number of related samples excludes a more detailed and quantitative analysis of this effect at this time.

The comparison in Fig. 7B between father and daughter also enables the study of the effect of hair care products, such as shampoo and conditioner on the molecular structure of hair. While Subject 2 (father) uses soap and shower gel to clean scalp and hair, Subject 1 (daughter) regularly uses shampoo and conditioner. The identical X-ray signals indicate that these products do not have an effect on the molecular structure of keratin and membranes deep inside the hair (within the resolution of our experiment).

We note that in order to maximize the scattered signals, the entire hair strand was illuminated in our experiments using a relatively large X-ray beam. Microbeam X-ray diffraction on synchrotron sources, which uses small, micrometre sized beams (Iida & Noma, 1993; Busson, Engstrom & Doucet, 1999; Kreplak et al., 2001b; Ohta et al., 2005; Kajiura et al., 2006), gives a high spatial resolution. By illuminating selective parts of the hair, the occurrence of the signals that we observed can be determined as a function of their location within the hair in future experiments.

Conclusions

We studied the molecular hair structure of several individuals using X-ray diffraction. Hair samples were collected from 12 healthy individuals of various characteristics, such as gender, optical appearance and genetic relation. Signals corresponding to the coiled-coil phase of the keratin molecules, the formation of intermediate filaments in the cortex and from the lipid molecules in the cell membrane complex were observed in the experiment. The corresponding signals were observed in all individuals, independent of gender or appearance of the hair, such as colour or waviness, within the resolution of this experiment. Given the small standard deviation of the molecular dimensions of these general features, anomalies possibly related to certain diseases should be easy to detect.

While all hair samples showed these general features, differences between individuals were observed in the composition of the plasma membrane in the cell membrane complex. Genetics seem to play an important role in the properties of these membranes, as genetically similar hair samples from father and daughter and identical twins showed identical patterns, though hair from fraternal twins did not.

Supplemental Information

Supplemental Information 1 Two-dimensional X-ray Data of All 12 Subjects

Two-dimensional X-ray data of all 12 subjects investigated in this study. Data are provided as 2-dimensional matrices in Matlab format (‘subject1.mat’). The file ‘PeerJ_load_data.m’ is a Matlab macro to load and visualize the 2-dimensional data sets.

Click here for additional data file.

Additional Information and Declarations

Competing Interests

Author Contributions

Human Ethics

The authors declare there are no competing interests.

Fei-Chi Yang, Yuchen Zhang and Maikel C. Rheinstädter conceived and designed the experiments, performed the experiments, analyzed the data, contributed reagents/materials/analysis tools, wrote the paper, prepared figures and/or tables, reviewed drafts of the paper.

The following information was supplied relating to ethical approvals (i.e., approving body and any reference numbers):

Hamilton Integrated Research Ethics Board (HIREB) under approval number 14-474-T.

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
