# Peer review of "The structure of people’s hair"

_PeerJ, doi:10.7717/peerj.619_

## Round 0.1 · original submission · Minor Revisions

Dear Authors,

We have received the reviewers' comments and the decision is minor revision.

·

Basic reporting

No comments

Experimental design

I have a few comments regarding the data collected and the interpretations provided:

- It is well known that the axial packing of coiled coils within keratin filaments in hair gives rise to a large number of fine arcs along the meridian (z axis for the authors). The most prominent one is at 6.7 nm and is not observed by the authors. However there are other ones between 6.7 and 1.2 nm. Instead of using an azimuthal integration with a 25 degree opening, the authors should integrate along the meridian with a thin rectangular box, a few pixel wide. I am pretty sure they will see a few peaks that should be reported for completeness.

- For both Figure 5 and 6, the authors fit the profiles with a few gaussian peaks plus a background signal that looks like a power law. The authors need to provide a detailed description of the fitting protocol and how initial values were chosen. Where all fitting parameters allowed to move freely over all possible values or restricted to a given window (this is a common practice for peak fitting)?
The issue of the power law background is important for fitting the SAXS data. The three peaks in the SAXS profile are quite broad and in truth not gaussian in nature as they derive from the peaks of a Bessel function. Their positions are easily shifted by changing the background function.

Validity of the findings

No Comments

Additional comments

Overall this is a well conducted study of the molecular structure of human hair over a small group of individuals. I find very interesting that most of the observed differences are in signals associated to lipids. In that regard the authors should mention possible lipid contribution in the medulla of human hair. Previous infrared spectroscopy maps of human hair indicated the predominance of the characteristic CH2 vibration in the medulla (Kreplak et al., Int J Cosmetic Sci, 2001, 23(6):369-74).

Reviewer 2 ·

Basic reporting

No comments.

Experimental design

No comments.

Validity of the findings

No comments.

Additional comments

The article reports on an x-ray fiber diffraction study of human hair. The authors use a laboratory x-ray source to record wide-angle diffraction patterns from fiber bundles. The study is based on a very small set of samples from 12 different individuals, as already pointed out in the abstract.

The authors list the measured hair diameters, color, and appearance in a table and qualitatively compare the recorded diffraction patterns. The features observed in the x-ray patterns are associated with kreatin proteins, fatty acids, and membrane components within the hierarchical hair structure.

As a main conclusion, the authors find that differences between the individuals were mainly observed in the composition of the plasma membrane, while all other features appear to be essentially equal among individuals.

The paper is of course not a systematic and statistically meaningful investigation due to the small number of individuals, and the paper does not claim to be. Also, more advanced methods such as x-ray microbeam diffraction on individual hairs are available. X-ray nanofocus techniques would even allow to study subdomains within an individual hair fiber. Nevertheless, the paper provides a useful and easy to read overview and possibly some inspiration for a more general audience. It is also novel in emphasizing the important role of lipids in the hair structure.

I would recommend that the authors explore in more detail what can be done with modern nanobeam diffraction methods and include this in the introduction and outlook, including relevant literature.

---

## Round 0.2 · accepted · Accept

We would be pleased if you could provide the raw data (as offered in previous correspondence) even if it is in some non-standard format.